# Prevalence and risk factors of work-related musculoskeletal disorders among shopkeepers in Ethiopia: Evidence from a workplace cross-sectional study

**Amensisa Hailu Tesfaye**[1]*, **Gebisa Guyasa Kabito**[1], **Fantu Mamo Aragaw**[2], **Tesfaye Hambisa Mekonnen**[1]

1 Department of Environmental and Occupational Health and Safety, Institute of Public Health, College of Medicine and Health Sciences, University of Gondar, Gondar, Ethiopia, 2 Department of Epidemiology and Biostatistics, Institute of Public Health, College of Medicine and Health Sciences, University of Gondar, Gondar, Ethiopia

* amensisahailu@gmail.com

**Data Availability Statement:** All relevant data are within the manuscript and its Supporting Information files.

## Abstract

### Introduction

Work-related musculoskeletal disorders (WRMSDs) are the leading cause of disability worldwide. Shopkeepers are prone to developing work-related musculoskeletal disorders, but they are largely overlooked in research and policy actions, particularly in developing countries. So far, there is a lack of data on the magnitude and factors influencing work-related musculoskeletal disorders among shopkeepers in Ethiopia. Therefore, the current study aimed to explore the prevalence and risk factors of work-related musculoskeletal disorders among shopkeepers in Gondar City, Ethiopia.

### Methods

A workplace-based cross-sectional study was conducted from July to August 2022, in Gondar city, Northwest Ethiopia. A multistage sampling technique was used to select 625 shopkeepers. The data were collected using an interviewer-administered standardized Nordic Musculoskeletal Questionnaire. Analysis was made using Stata version 14. Factors associated with the prevalence of work-related musculoskeletal disorders were identified using the multivariable Poisson regression model. The adjusted prevalence ratio with 95% confidence intervals (CIs) and p-value < 0.05 were applied to establish the significance of associations.

### Results

The overall prevalence of work-related musculoskeletal disorders among shopkeepers in the past 12 months was found to be 81.1% (N = 507). The most frequently affected body part was the lower back (46.6%), followed by the upper back (43.8%) and shoulder (35.4%). Being female ($p = 0.043$), being in the age group of $\geq$40 years ($p = 0.028$), being overweight ($p = 0.035$), experiencing job stress ($p = 0.006$) and prolonged sitting ($p = 0.045$) were

**Funding:** The author(s) received no specific funding for this work.

**Competing interests:** The authors have declared that no competing interests exist.

**Abbreviations:** AOR, Adjusted Odds Ratios; BMI, Body Mass Index; CI, Confidence Interval; COR, Crude Odds Ratio; ETB, Ethiopian Birr; Epi Info, statistical software for epidemiology developed; MSDs, Musculoskeletal Disorders; NMQ, Nordic Musculoskeletal Questionnaire; OR, Odds Ratios; STATA, Statistical software for data science; SDGs, Sustainable Development Goals; WRMSDs, Work-related Musculoskeletal Disorders; YLDs, Years Lived with Disability.

significant factors for the prevalence of work-related musculoskeletal disorders among shopkeepers.

## Conclusion

This study revealed that shopkeepers face an alarmingly high prevalence of work-related musculoskeletal disorders. Female, older, overweight, stressed and shopkeepers who sit in the same position for long periods of time were identified as particularly vulnerable groups. These findings call for the urgent development and implementation of preventive measures, including ergonomic adjustments, education and training programs, stress management techniques and the promotion of physical activity, to protect this vulnerable workforce from the debilitating effects of work-related musculoskeletal disorders and to ensure their long-term health and well-being.

## Introduction

Work-related musculoskeletal disorders (WRMSDs) are a group of painful conditions that affect the musculoskeletal system, including muscles, nerves, tendons, joints, cartilages, ligaments, spinal discs, bones and related soft tissues. These disorders are directly related to workplace risk factors and are commonly attributed to occupational activities and conditions [1–3]. In low- and middle-income countries, the rate of WRMSDs is quite high due to poor working conditions and a lack of effective work-related injury prevention and control programs [4]. Work-related musculoskeletal disorders can progress from mild to severe forms over time and are seldom life-threatening but reduce the quality of life for a large proportion of the world's working population [1].

Musculoskeletal disorders (MSDs) caused by occupational exposures have become major health problems around the world. [5]. Globally, WRMSDs are a leading contributor to work-related injury and disability [6, 7]. They have become a global public health concern [8, 9] due to their overall consequences, which include decreased productivity [10, 11], work absenteeism [12], limited mobility, resulting in early retirement from work, reduced quality of life, physical disability, and mental disorders [1, 7, 13, 14]. The Global Burden of Disease (GBD) 2019 data shows that around 1.71 billion people worldwide are living with MSDs, and these disorders are the largest contributor to years lived with disability (YLD) worldwide, accounting for around 149 million YLDs, or 17% of all YLDs worldwide [15]. The burden of MSD is increasing across the entire Sub-Saharan population [16]. WRMSDs and other musculoskeletal conditions are less prioritized and empirically underrepresented in low and middle-income countries (LMICs), particularly in Ethiopia, due to a focus on more pressing and life-threatening health issues such as poverty and infectious diseases [17].

According to the latest estimates from the Labour Force Survey (LFS), 470,000 workers were affected by WRMSDs in 2020/21. This represents a prevalence rate of 1,420 per 100,000 workers and accounts for 28.0% of all work-related ill health [1]. As per reports from Canada and the Netherlands [18, 19], the prevalence of MSDs ranges from 29.0% to 74.5%. A high prevalence of WRMDs has been recorded among workers who are exposed to manual labor, work in unusual and restricted postures, repetitive and static work, vibrations, and poor psychological and social conditions [5, 20]. Shopkeepers work in physically demanding and perilous working conditions, which lead to the development of WRMSDs [21]. Studies conducted

in Asia reported a prevalence of low back pain of 57.3% [21] and 56.3% [22]. Shopkeepers experienced severe pain 30.0% and work difficulties 58.0% [22], musculoskeletal pain 70.6% and knee pain 30.0% [21]. WRMSDs are a problem in all African countries, with the prevalence of the musculoskeletal disease ranging from 15.0% to 93.6% [23]. In Ethiopia, the prevalence of WRMSDs ranges from 35.0% to 74.5% according to a few studies conducted in different parts of the working population [24]. However, studies on WRMSDs among shopkeepers are lacking.

Shopkeepers are people who run a shop as their sole source of income [21]. They frequently work long and erratic hours, including at night and on weekends [25]. Thus, due to the nature of the shopping industry, shopkeepers are regularly vulnerable to risk factors such as prolonged standing, frequent lifting, awkward posture, and constant carrying of stock that are associated with the involvement of WRMSDs [21, 22]. An extended sitting position is also common when customers are not visiting shopping centers, potentially increasing their risk of developing musculoskeletal conditions [21]. The occurrences of WRMSDs are also influenced by personal or individual factors [5, 26]. Depending on the previous investigations, these factors include age, gender, alcohol/tobacco consumption, physical activity, endurance, anthropometry, and chronic illness, as well as an educational level [5, 26–30]. Furthermore, psychosocial risk factors related to workplace emotional perception influence the risk of developing WRMSDs [4, 26, 31].

Evidence supporting labor rights and promoting safe working conditions is scarce at a global level. Shopkeepers continue to work in dangerous conditions that jeopardize their health, safety and well-being. They are at a high risk of developing WRMSDs, but they are largely overlooked in research and policy actions, particularly in developing countries. In Ethiopia, minimal attention has been paid to implementing health and safety programs in the informal economies, particularly in business sectors such as the shopping industries. So far, there is a lack of data on the magnitude and factors influencing WRMSDs among shopkeepers in Ethiopia. Therefore, the current study aimed to determine the prevalence and associated factors of WRMSDs among shopkeepers in Gondar City, Ethiopia. The findings of this study may contribute to the achievement of the United Nations 2030 Agenda of Sustainable Development Goals (SDGs), specifically SDG 8.8 (protect labor rights and promote safe and secure working environments for all workers). It also provides relevant data for policymakers and concerned stakeholders to design and implement prevention and control measures to help alleviate further episodes.

## Materials and methods

### Study design, period, and setting

A workplace-based cross-sectional study was conducted from July to August 2022 in Gondar City. The city is located 727 km away from Addis Ababa, the capital of Ethiopia. Most the city's economic activities are centered on trading, and many city residents are involved in the trading sector to generate their income. Shopping accounted for a larger portion of the city's total business [32]. According to the Gondar city department of trade and market development report, there were 10,432 shops and 17,648 shopkeepers working in Gondar city at the time of the study.

### Source and study populations

All shopkeepers in Gondar city were the source population, whereas the randomly selected shopkeepers working in selected sub-cities were the study population. Shopkeepers who had worked for at least 12 months before the study were eligible for this study, while those who

had a history of accidents involving the musculoskeletal system (fractures of the spine or limbs due to a car accident or fall), spinal surgeries, and major surgeries on any part of the body, congenital anomalies such as spinal and limb anomalies, and pregnant women were excluded because they could potentially bias the results of the study [33, 34].

## Sample size determination and sampling procedure

The sample size was calculated using a single population proportion formula [35]:

$$n = \left(Z\alpha/_2\right)^2 \frac{[p(1-p)]}{d^2}$$

Where: n = initial sample size

Zα/2 = the value of the normal distribution at α/2, for α is 0.05 the value of z is 1.96

p = proportion of WRMSDs (50%)

d = tolerance error (5%)

$$n = \left(1.96\right)^2 \frac{[0.5(1-0.5)]}{0.05^2} = 384.$$

Then, using a design effect of 1.5 and 10% non-response rate, the final sample size raised to *n* = 634 [36].

Two sub-cities, Maraki and Arada, were selected using the lottery method. The sample was proportionally allocated to each sub-city and shops were selected using a systematic random sampling technique. Shops in the Maraki sub-city were selected every four shops, while shops in the Arada sub-city were selected every five shops. The first shop in each sub-city was selected using the lottery method (**Fig 1**). If two or more shopkeepers were found in a shop, one of them was selected by lottery.

## Operational definitions

**Work-related musculoskeletal disorders (WRMSDs):** a self-reported pain, ache, or discomfort in any part of the neck, shoulder, upper back, lower back, hip/thigh, knee/leg, ankle/foot, or wrist/hand at any time in the previous 12 months [4, 37, 38].

**High perceived severity:** a pain intensity score of 50 or greater and less than 3 disability points [39]

**Low perceived severity:** a pain intensity score of less than 50 and less than 3 disability points [39].

High perceived disability: a pain disability point score of 3–6 points [39].

**Low perceived disability:** a pain disability point score of less than 3 points [39].

**Perceived job stress:** a score of at least 21 on the workplace-stress scale [40].

**Perceived job satisfaction:** the total score of at least 32 on the generic job satisfaction scale [41].

**Body mass index:** weight in kilograms divided by the square of the height in meters (kg/m2) categorized as underweight = body mass index (BMI) <18.50, normal (health) = BMI 18.50–24.99, overweight/obese = BMI ≥25.00 [4].

**Physically active:** exercising or doing any kind of sports activity at least two times per week with a duration of at least 30 minutes [4, 42].

**Cigarette smoking:** It is the practice of smoking cigarette by shopkeepers for at least one stick of cigarette per day [43].

**Alcohol intake:** the consumption of any kind of alcohol at least two times per week [43].

**Khat chewer:** chewing khat three times a week for at least 12 months [44, 45].

**Awkward postures:** working for two or more hours per day with the neck bent without support; working with a bent wrist; working with the back bent without support; squatting and kneeling [5].

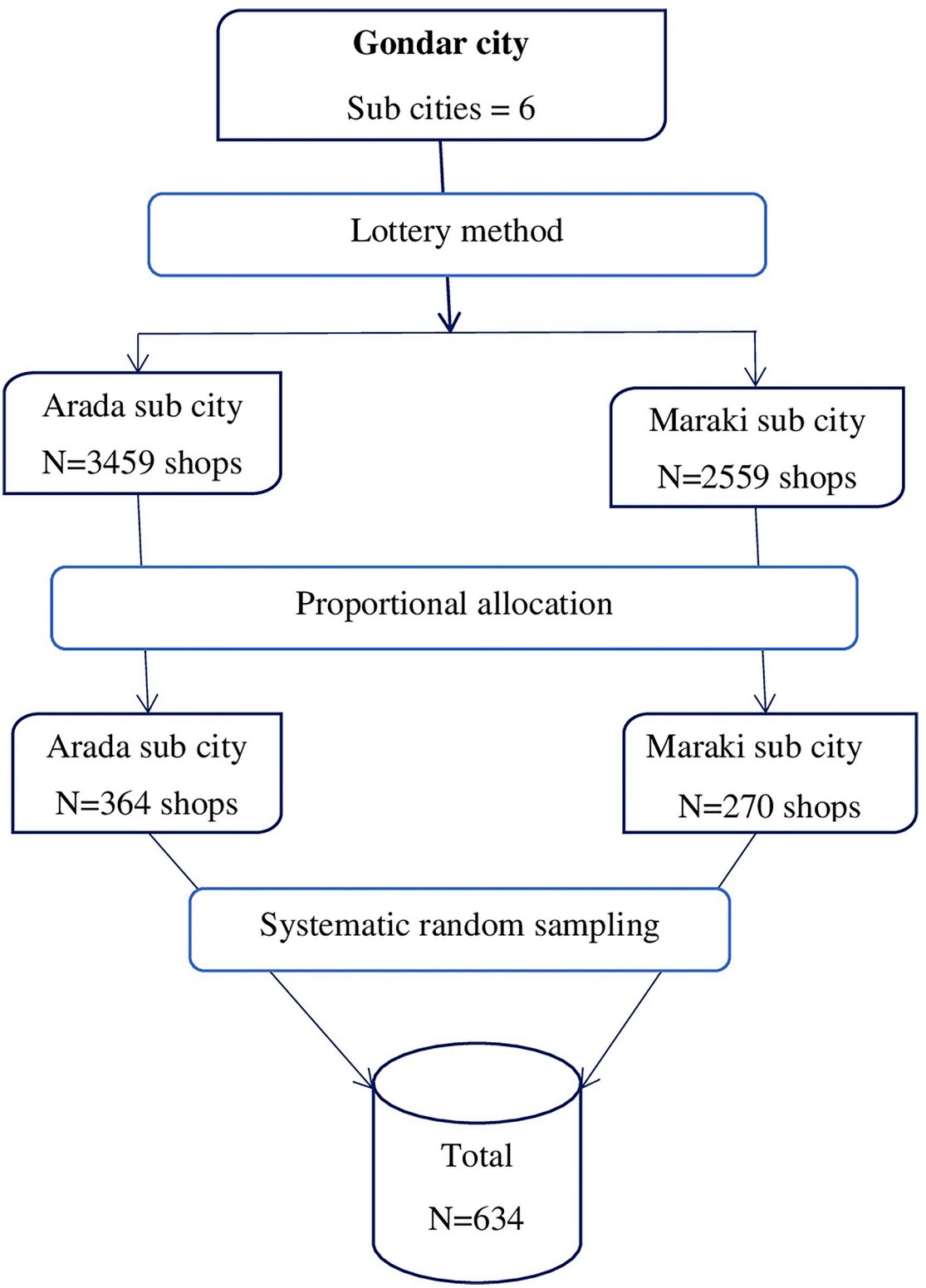

**Fig 1. Schematic representation of sampling procedure for shopkeepers in Gondar city, northwest Ethiopia.**

**Adjustable sitting chairs:** chairs have wheels or castors suitable for the floor surface, have adjustable seat height, and are available within a distance of at least 1.5 meters [46].

**Take rest breaks:** take brief rest breaks every 60 to 120 minutes. During these breaks, stand up, move around, and do something else, or get a soft drink or a cup of coffee or tea [4].

**Repetitive work:** repeated the same motion for less than 30 seconds or no variation every few seconds for two or more hours per day [47].

**Static posture:** sitting or standing in a restricted space for two or more hours without changing positions [47].

**Chronic illness:** self-reported illnesses such as asthma, diabetes mellitus, stroke, kidney stone, and hypertension that can be managed but not cured [48].

## Data collection tools and procedures

Data were collected using a standardized Nordic Musculoskeletal Questionnaire (NMQ) tool [37]. The questionnaire was divided into four sections. The first section was designed to collect socio-demographic data such as age, gender, educational status, marital status, religion, years of current work experience, monthly income, and family size of respondents. The second section of the questionnaire is focused on the assessment of work-related musculoskeletal disorders and was collected using NMQ. The NMQ is recommended for use in epidemiological and work-related musculoskeletal disorders surveys as well as in consultations for musculoskeletal disorders and ergonomic-related problem assessment. The NMQ has been used to measure the magnitude of pain, ache, or discomfort in the neck, shoulder, upper back, lower back, hip/thigh, knee/leg, ankle/foot, wrist/hand, and elbow [37]. The NMQ has been validated and proven to be reliable in a variety of languages with acceptable psychometric properties [49, 50]. In Ethiopia, previous studies have made extensive use of the NMQ to investigate the public health problems associated with disorders in different populations [5, 28, 51]. The questionnaire's third section contains information used to assess behavioral and psychosocial factors such as cigarette BMI (kg/m$^2$), physical activity (yes/no), smoking (yes/no), alcohol consumption (yes/no), chewing khat (yes/no), history of chronic illness (yes/no), perceived severity and disability of musculoskeletal pains, as well as perceived job satisfaction and perceived job stress.

The 7-items Von korff et al. [39] the questionnaire was used to assess the shopkeeper's perceived severity and disability of musculoskeletal pain. This instrument is graded on a scale of 0 to 10 and has a total score of 100. The perceived severity of pain was calculated by the formula = (((response question 1) + (response question 2) + (response question 3)) / 3) * 10 and perceived disability score = (((response question 5) + (response question 6) + (response question 7)) / 3) * 10 and disability points = (points for disability days or question number 4) + (points for disability score). A final score was categorized into two with a score of <50 or disability points 3 = 0 (low perceived severity), a score of ≥50 or disability points < 3 = 1 (high perceived severity), and perceived disability = disability points < 3 = 0 (low perceived disability) and disability points ≥ 3 = 1 (high perceived disability).

To assess shopkeeper job satisfaction, a 10-item generic job satisfaction scale questionnaire was used [41]. The scale comprised ten questions ranging from 1 to 5 for each item and ranged from very dissatisfied, dissatisfied, neutral, satisfied, and very satisfied, according to their occurrence respectively. The scale had 10 items rated from 1 to 5, and the responses ranged from very dissatisfied, dissatisfied, neutral, satisfied and very satisfied, depending on how often they occurred in the month prior to the survey and then summarized across all 10 items. The scale produced a single ranking, with high scores indicating higher job satisfaction and vice versa. Perceived job-related stress of the participants was collected using the 8-item

workplace stress scale questionnaire [40]. The scale comprised eight questions ranging from 1 to 5 for each item and ranged from never, rarely, sometimes, often, and very often, according to their occurrence respectively. The 8-item workplace stress scores are obtained by reversing scores on three positive items, e.g., 5 = 1, 4 = 2, 3 = 3, etc., and then summing up all 8 items. Items 6, 7, and 8 are positive items. The scale produced a single ranking, with high scores indicated higher stress levels and vice versa. All instruments used in the current study have been employed in previous studies conducted in the country's context [4, 5, 29].

The final section of the tool incorporates work-related environment and ergonomics characteristics of study participants, including location of shop, type of shop, working hours (per day and peer week), working postures (sitting, standing, awkward posture, and repetitive work), the practice of taking breaks, and type of sitting chair (**S1 File**).

## Data quality control

The questionnaire was first prepared in English by the investigators and then translated into Amharic and back into English by language experts and physiotherapists to ensure consistency. We recruited three final year environmental and occupational health and safety students for face-to-face data collection and an environmental health professional as supervisor. A one-day training was given to the data collectors and supervisor on topics related to research objectives, the clarity of the data collection tool, techniques of interviewing, research ethics (the confidentiality of information and consent in the study), and selection methods of study participants. The training was delivered in lecture and discussion formats. The questionnaires were pre-tested on 5% of the sample size (32 shopkeepers) that were not included in the final analysis, and the relevant modifications were made one week before the actual data collection. The average amount of time that it took for an interview to be completed was 40 minutes. Problems encountered during the data collection process were resolved through on-the-spot discussions with the principal investigator, supervisor, and data collectors. Besides, the principal invigorator supervised the interview process. The investigators double-checked the completeness and accuracy of all completed questionnaires daily.

## Data management and statistical analyses

The collected data were manually entered into Epi Info version 7 software for cleaning, coding, and checking for missing values and exported into Stata version 14 software for further analysis. Descriptive statistics were computed and presented using frequencies, percentages, means, standard deviations, tables, and graphs. Multicollinearity among independent variables was assessed using variance inflation factors (VIF) and found to be acceptable (all variables had VIF <5 values) [52]. The reliability of the questionnaire was also tested using Cronbach's alpha ($\alpha$). The Cronbach's alpha ($\alpha$) value for the NMQ was 0.76. The Cronbach's alpha values were 0.75 for pain intensity and 0.73 for disability. The Cronbach's alpha ($\alpha$) value for the 10-item Job Satisfaction Scale questionnaire was 0.79, and the Cronbach's alpha ($\alpha$) value for the 8-item Workplace Stress Scale questionnaire was found to be 0.82. As a result, the internal consistency of the questionnaire was found to be acceptable and satisfactory [53]. The Poisson regression model with robust variance was used to investigate the statistical association between the prevalence of work-related musculoskeletal disorders and predictor variables [54]. The measure of effect was the prevalence ratio (PR) with 95% confidence intervals (CIs). To control for the effects of potential confounders, independent variables with a significance level of $p < 0.20$ in the bivariable analysis were included in a Poisson regression model with robust variance. The adjusted prevalence ratio (PR) with 95% confidence intervals (CIs) and p-value $< 0.05$ were applied to establish the significance of associations in the multivariable Poisson regression model.

### Ethical approval and consent to participate

Ethical clearance was obtained from the Institutional Ethical Review Board (IERB) of the University of Gondar (**Reference No: EOHS/IRBN/639//2022**). All methods were carried out following the Helsinki Declaration and the guidelines [55] and regulations of the University of Gondar's research ethics review committee. All study participants were informed about the purpose of the study and the importance of their participation in their local language. Thus, the information sheet that clearly shows the research topic, the objectives of the study, confidentiality of the participant's responses, the study benefits, and associated risks was prepared. The information sheet and consent were provided for respondents to read for those who could read, while the interviewer read the consent form for those who could not read. The respondents were then asked if they were willing to take part in the study, and their written informed consent was obtained. Once they had agreed to participate in the study, the face-to-face interview was conducted to preserve their ability to carry out their normal business activities, as they were busy at work to fill out the administered questionnaires themselves. They were also allowed to ask any questions they had about the research and to reject or terminate the interview at any time. Any personal identifiers were eliminated to ascertain confidentiality and only anonymous data were used for interpretations. The collected data were entered and analyzed solely on the principal investigator's computer and were password-protected. Respondents who complained of severe musculoskeletal disorders during the data collection period were advised to look for medical assistance.

## Results

### Socio-demographic characteristics of study participants

A total of 634 shopkeepers were invited to participate in the study, but data from 9 shopkeepers were excluded from further analysis due to incomplete data. Complete data from 625 participants were analyzed, giving a response rate of 98.6%. About half (51.4%) of the respondents were male and the mean (±SD) age was 36.7 (± 11.5) years. Three hundred and seventy-five (60.0%) were married, and 223 (35.7%) had work experience of 6–10 years (**Table 1**).

### Behavioral and psychosocial characteristics of study participants

The majority of shopkeepers, 443 (70.9%), had a normal BMI (18.5–24.9 kg/m$^2$), while 151 (24.2%) had an overweight BMI (> 18.5 kg/m$^2$). Regarding physical activity, only 73 (11.7%) of those surveyed engaged in any physical exercise. Fifty-six (9%) of them have cigarette smoking behaviors, and 181 (29%) of them consume alcohol at least twice a week. Eighty (12.8%) of shopkeepers have a medical history of chronic illnesses such as asthma (n = 26), hypertension (n = 15), diabetes mellitus (n = 33), heart disease (n = 2), and kidney stones (n = 4). The majority, 498 (79.7%) of the shopkeepers, had a low perceived severity of musculoskeletal pain, while 127 (20.7%) had a high perceived severity of it. Around three-fourths of the shopkeepers (74.4%) had a low perceived disability of musculoskeletal pain, whereas about one-fourth of them (25.6%) had a high perceived severity of the pain. In terms of psychosocial characteristics, 173 (27.7%) of the shopkeepers were dissatisfied with their current job, and 401 (64.2%) of them described being perceived to be stressed because of their job (**Table 2**).

### Work-related environment and ergonomic characteristics of study participants

Small local shops (wholesalers) are the most common type of shop in the study (30.7%), and the majority of shopkeepers (84.5%) work on the ground floor. The majority, 447 (71.5%) of

**Table 1. Socio-demographic characteristics of shopkeepers in Gondar City, Ethiopia, 2022 (N = 625).**

| Variables | Frequency | Percent (%) |
|---|---|---|
| **Gender** | | |
| Male | 321 | 51.4 |
| Female | 304 | 48.6 |
| **Religion** | | |
| Orthodox | 338 | 54.1 |
| Muslim | 178 | 28.5 |
| Protestant | 94 | 15.0 |
| Catholic | 15 | 2.4 |
| **Age in years** | | |
| 20–29 | 196 | 31.40 |
| 30–39 | 228 | 36.50 |
| ≥40 | 201 | 32.10 |
| Mean ±SD = 36.7 ± 11.5 | | |
| **Marital status** | | |
| Single | 178 | 28.5 |
| Married | 375 | 60.0 |
| Divorced | 34 | 5.4 |
| Widowed | 38 | 6.1 |
| **Educational level** | | |
| Unable to read and write | 13 | 2.1 |
| Primary school complete | 250 | 40.0 |
| Secondary school complete | 265 | 42.4 |
| Certificate/diploma/degree | 97 | 15.5 |
| **Work experience in current shop (years)** | | |
| 1–5 | 304 | 48.6 |
| 6–10 | 223 | 35.7 |
| >10 | 98 | 15.7 |
| **Monthly salary (ETB)** | | |
| <4000 | 218 | 34.9 |
| ≥4000 | 407 | 65.1 |
| **Family size in person** | | |
| <5 | 209 | 33.4 |
| ≥5 | 416 | 66.6 |

**Key:** ETB = Ethiopian Birr (currency)

the participants worked more than 8 hours per day at their job, and more than half, 346 (55.4%) of them worked more than five days per week. More than two-thirds (61.9%) of shopkeepers work for two or more hours per day with a bent neck without support, a bent wrist, a bent back without support, squatting, and kneeling. Four hundred and thirty-nine (70.2%) of the shopkeepers explained that they had not taken any breaks at their workplace. Only 7.5% of those who responded to the survey had used adjustable sitting chairs. The majority of shopkeepers, 62.6% and 58.4%, respectively, sat and stood in a confined space for two or more hours per day without changing position. The majority of respondents, 393 (62.9%), stated that their job required repetitive movements (**Table 3**).

**Table 2. Behavioral and psychosocial characteristics of shopkeepers in Gondar City, Ethiopia, 2022 (N = 625).**

| Variables | Frequency | Percent (%) |
|---|---|---|
| **Body mass index (BMI)** | | |
| Underweight | 31 | 5.0 |
| Normal | 443 | 70.9 |
| Overweight | 151 | 24.2 |
| **Physical exercise** | | |
| Yes | 73 | 11.7 |
| No | 552 | 88.3 |
| **Cigarette smoke** | | |
| Yes | 56 | 9.0 |
| No | 569 | 91.0 |
| **Drink alcohol** | | |
| Yes | 181 | 29.0 |
| No | 444 | 71.0 |
| **Khat chawing behavior** | | |
| Yes | 48 | 7.7 |
| No | 577 | 92.3 |
| **Chronic illness** | | |
| Yes | 80 | 12.8 |
| No | 545 | 87.2 |
| **Chronic illness reported** | | |
| Asthma | 26 | 4.2 |
| Hypertension | 15 | 2.4 |
| Diabetes | 33 | 5.3 |
| Others[#] | 6 | 1.0 |
| **Perceived musculoskeletal pain severity** | | |
| High | 127 | 20.3 |
| Low | 498 | 79.7 |
| **Perceived musculoskeletal pain disability** | | |
| High | 160 | 25.6 |
| Low | 465 | 74.4 |
| **Perceived job satisfaction** | | |
| Satisfied | 452 | 72.3 |
| Not satisfied | 173 | 27.7 |
| **Perceived job stress** | | |
| Stressed | 401 | 64.2 |
| Not stressed | 224 | 35.8 |

Key

[#]Heart disease, Kidney stone

## Prevalence of work-related musculoskeletal disorders among shopkeepers

This study revealed that the overall prevalence of self-reported one or more work-related musculoskeletal disorders among shopkeepers in northern Ethiopia during the previous 12 months was 81.1% (n = 507) [95% CI (77.8% to 84.1%)], and the previous 7-days prevalence was 75.2% (n = 470) [95% CI (71.6% to 78.5%)]. Lower back pain (46.6%), upper back pain (43.8%), and shoulder pain (35.4%) were the most frequently reported complaints among work-related musculoskeletal disorders in the previous 12 months (**Figs 2 and 3**). Furthermore, among

**Table 3. Work-environment and ergonomic characteristics of shopkeepers in Gondar City, Ethiopia, 2022 (N = 625).**

| Variables | Frequency | Percent (%) |
|---|---|---|
| **Workplace (shop) location** | | |
| Ground | 528 | 84.5 |
| First floor | 62 | 9.9 |
| Second floor and above | 35 | 5.6 |
| **Type of shop** | | |
| Supermarket | 35 | 5.6 |
| Clothe, shoe, and bag shop | 162 | 25.9 |
| Small local shop (wholesalers) | 192 | 30.7 |
| Stationery and bookstore shop | 64 | 10.2 |
| Electronics and hardware shop | 97 | 15.5 |
| Greengrocer (fruit and vegetable) shop | 56 | 9.0 |
| Others[b] | 19 | 3.0 |
| **Working hours per day** | | |
| ≤8hrs | 178 | 28.5 |
| >8hrs | 447 | 71.5 |
| **Total working day per week** | | |
| ≤5 days/week | 279 | 44.6 |
| >5 day/week | 346 | 55.4 |
| **Awkward postures** | | |
| Yes | 387 | 61.9 |
| No | 238 | 38.1 |
| **Taking rest breaks** | | |
| Yes | 186 | 29.8 |
| No | 439 | 70.2 |
| **Types of sitting chair** | | |
| Fixed | 578 | 92.5 |
| Adjustable | 47 | 7.5 |
| **Prolonged standing** | | |
| Yes | 391 | 62.6 |
| No | 234 | 37.4 |
| **Prolonged sitting** | | |
| Yes | 260 | 41.6 |
| No | 365 | 58.4 |
| **Repetitive movement** | | |
| Yes | 393 | 62.9 |
| No | 232 | 37.1 |
| **Lifting or carrying heavy loads** | | |
| Yes | 369 | 59.0 |
| No | 256 | 41.0 |
| **Climbing stairs or ladders** | | |
| Yes | 328 | 52.5 |
| No | 297 | 47.5 |

Key

[b]souvenir, jeweler, and cosmetics shop

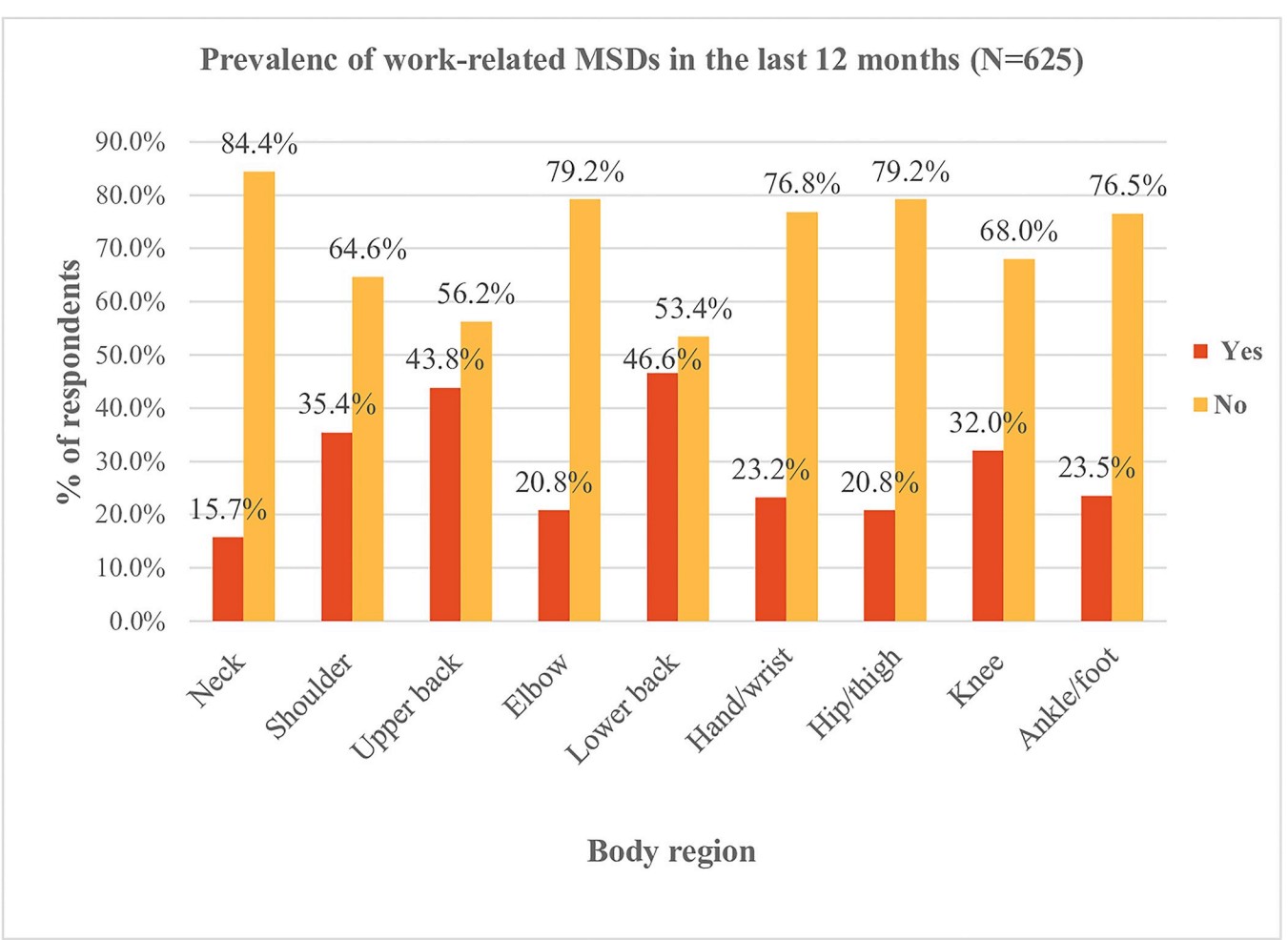

**Fig 2. Prevalence of work-related musculoskeletal disorders by body regions among shopkeepers in the last 12 months.**

work-related musculoskeletal disorders, lower back pain and upper back pain showed a significant difference in prevalence between males and females (**Table 4**).

## Factors associated with work-related musculoskeletal disorders

Multivariable robust Poisson regression analysis was used to determine factors associated with the prevalence of work-related musculoskeletal disorders. In adjusted Poisson regression analysis sex, age, body mass index (BMI), perceived job stress, and prolonged sitting in a confined space for two or more hours per day without changing position were significantly associated with work-related musculoskeletal disorders.

Female shopkeepers had 1.11 times the prevalence of work-related musculoskeletal disorders compared with male shopkeepers (PR = 1.11; 95%CI: 1.01–1.33), with a p-value of 0.043. The age of the participants was also found to be significantly associated with the occurrence of work-related musculoskeletal disorders. The prevalence of work-related musculoskeletal disorders is 23% higher among shopkeepers aged ≥40 years compared with those aged 20–29 years (PR = 1.23; 95%CI: 1.02–1.56), with a p-value of 0.028. The prevalence of work-related musculoskeletal disorders was 1.09 times higher in shopkeepers with an overweight body mass index (≥25 kg/m2) compared with those with a normal body mass index (18.5–24.9 kg/m$^2$)

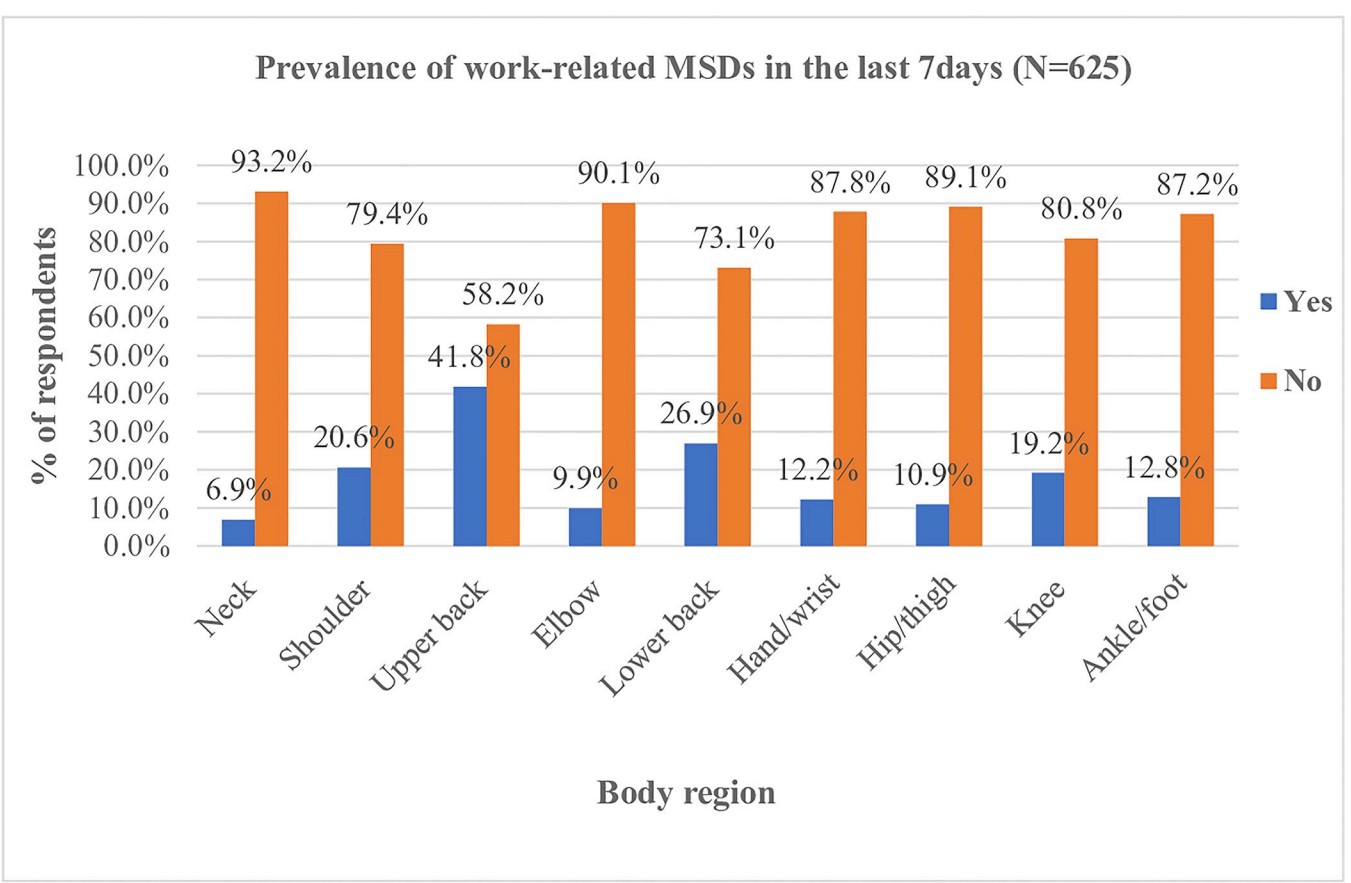

**Fig 3. Prevalence of work-related musculoskeletal disorders by body regions among shopkeepers in the last 7 days.**

(PR = 1.09; 95%CI: 1.10–1.33), with a p-value of 0.035. In addition, shopkeepers experiencing job stress had 1.32 times the prevalence of work-related musculoskeletal disorders compared with non-stressed shopkeepers (PR = 1.32; 95%CI: 1.08–1.60), with a p-value of 0.006. Finally, multivariable analysis of this study displayed that the prevalence of work-related musculoskeletal disorders was 18% higher among shopkeepers who sat for long periods at their workplace compared with those who did not (PR = 1.18; 95%CI: 1.06–2.37), with a p-value of 0.045 as shown in Table 5.

## Discussion

In this study, the prevalence and associated factors of work-related musculoskeletal disorders among shopkeepers were determined. The finding shows the overall prevalence of work-related musculoskeletal disorders among the shopkeepers during the previous 12 months was 81.1%. The prevalence during the past 7 days has been reported in 75.2% of the participants. This finding highlights that exposure to workplace factors that are associated with the experience of work-related musculoskeletal conditions among shopkeepers is prominent. This implies that in Ethiopia, minimal attention has been given to implementing health and safety programs in the informal economies, notably in the business sectors, such as the shopping industries.

The current study showed that the low back is the most frequently affected body part (46.6%). The reports in India (57.33%) [21] and Pakistan (56.25%) [22], however, documented

**Table 4.  The 12 months prevalence of work-related musculoskeletal disorders by body region (N = 625).**

| Work-related musculoskeletal disorders | | Male | Female | χ2 value | p-value |
|---|---|---|---|---|---|
| | | Frequency (%) | Frequency (%) | | |
| Neck | Yes | 48 (15%) | 50 (16.4%) | 0.2636 | 0.608 |
| | No | 273 (85%) | 254 (83.6%) | | |
| Shoulder | Yes | 110 (34.3%) | 111 (36.5%) | 0.3444 | 0.557 |
| | No | 211 (65.7%) | 193 (63.5%) | | |
| Upper back | Yes | 115 (35.8%) | 167 (54.9%) | 16.6823 | **0.000**\* |
| | No | 206 (64.2%) | 137 (45.1%) | | |
| Elbow | Yes | 69 (21.5%) | 61 (20.1%) | 0.1937 | 0.660 |
| | No | 252 (78.5%) | 243 (79.9%) | | |
| Lower back | Yes | 124 (38.6%) | 167 (54.9%) | 16.6823 | **0.000**\* |
| | No | 197 (61.4%) | 137 (45.1%) | | |
| Hand/wrist | Yes | 75 (23.4%) | 70 (23.0%) | 0.0100 | 0.920 |
| | No | 246 (76.6%) | 234 (77.0%) | | |
| Hip/thigh | Yes | 63 (19.6%) | 67 (22.0%) | 0.5520 | 0.458 |
| | No | 258 (80.4%) | 237 (78.0%) | | |
| Knee | Yes | 110 (34.3%) | 90 (29.6%) | 1.5599 | 0.212 |
| | No | 211 (65.7%) | 214 (70.4%) | | |
| Ankle/foot | Yes | 74 (23.1%) | 73 (24.0%) | 0.0800 | 0.777 |
| | No | 247 (76.9%) | 231 (76.0%) | | |

Chi-square Test

\*Significant p-value $< 0.001$

a higher prevalence compared to our findings. The data from Pakistan explicitly focused on low back pain, whereas our survey assessed the overall prevalence as the primary objective, and the prevalence from India was assessed over the 3 months duration, which could explain the contradictory findings. Whereas 43.8% and 35.4% of the shopkeepers in our survey indicated they suffered from upper back and shoulder pain, respectively, which is higher than the results from the study in India (10% versus 15%). Such heterogeneity is most likely caused by differences in sample size.

Our survey showed a knee pain prevalence of 32%, which is comparable to the prevalence rate of 30% in India [21]. The nature of the activities associated with shop keeping, such as prolonged sitting and standing in a static position, which may result in the development of muscle and skeletal system dysfunction is almost similar everywhere. Besides, measures to prevent injuries and to safeguard the health and safety of workers are often lean in many developing nations [56–58].

In the multivariable analysis, we found a significant association between sex and the experience of WRMSDs among shopkeepers. Hence, females had a greater chance of developing WRMSDs. This result is supported by previous studies in Ethiopia [34], Ghana [59], India [60], and China [61]. This could be due to physiological, morphological, or sociocultural differences between males and females. Females have less muscle strength and a higher percentage of type 1 muscle fibers (weaker in nature) than males of the same size, making them more prone to MSDs [62, 63]. Moreover, most work environments (e.g., surface height, tool design, equipment size) are usually designed to fit anthropometric dimensions and the strength of a male's capabilities, while females are also engaged in the same work setup, which may place additional strain on female bodies. Furthermore, females' home-work interface, such as handling various household-related tasks (cleaning, cooking, childcare, laundry, etc.), may put

**Table 5. Association (χ2) and prevalence ratio with 95% confidence intervals based on the multivariable robust Poisson regression analysis for factors associated with work-related musculoskeletal disorders among shopkeepers in Gondar City, Ethiopia, 2022 (N = 625).**

| Variables | Work-Related Musculoskeletal Disorders | | PR | 95%CI | p-value |
|---|---|---|---|---|---|
| | **Yes** | **No** | | | |
| **Sex** | | | | | |
| Male | 250 | 71 | 1 | | |
| Female | 257 | 47 | **1.11** | **1.01–1.33** | **0.043*** |
| **Age** | | | | | |
| 20–29 | 139 | 57 | 1 | | |
| 30–39 | 187 | 41 | 1.13 | 0.90–1.42 | 0.288 |
| ≥40 | 181 | 20 | **1.23** | **1.02–1.56** | **0.028*** |
| **Work experience in years** | | | | | |
| 1–5 | 240 | 64 | 1 | | |
| 6–10 | 181 | 42 | 1.07 | 0.88–1.31 | 0.487 |
| >10 | 86 | 12 | 1.06 | 0.82–1.38 | 0.632 |
| **Family size** | | | | | |
| <5 | 163 | 46 | 1 | | |
| ≥5 | 344 | 72 | 1.28 | 0.85–1.26 | 0.751 |
| **Body mass index (BMI)** | | | | | |
| Underweight | 20 | 11 | 0.90 | 0.57–1.43 | 0.658 |
| Overweight | 136 | 15 | **1.09** | **1.10–1.33** | **0.035*** |
| Normal | 351 | 92 | 1 | | |
| **Chronic illness** | | | | | |
| Yes | 70 | 10 | 1.03 | 0.69–1.54 | 0.867 |
| No | 437 | 108 | 1 | | |
| **Perceived job stress** | | | | | |
| Stressed | 361 | 40 | **1.32** | **1.08–1.60** | **0.006*** |
| Not stressed | 146 | 78 | 1 | | |
| **Perceived musculoskeletal pain severity** | | | | | |
| High | 109 | 18 | 1.05 | 0.75–1.46 | 0.774 |
| Low | 398 | 100 | 1 | | |
| **Prolonged sitting** | | | | | |
| Yes | 224 | 36 | **1.18** | **1.06–2.37** | **0.045 *** |
| No | 283 | 82 | 1 | | |

**Keys:** 1 = reference category, PR: prevalence ratio; CI: confidence interval * = Significant association (P < 0.05)

additional physical and psychological burdens on them, aggravating the probability of having WRMSDs more than males [64].

Our analysis outlines that the prevalence of WRMSDs is 1.23 times higher among shopkeepers aged ≥40 years than among those aged 20–29 years. This finding corroborates other studies [5, 65, 66]. This could be explained musculoskeletal tissue susceptibility to bone fragility, loss of cartilage resilience, reduced ligament elasticity, loss of muscle strength, and fat redistribution, and reducing the ability of the tissues to carry out their normal functions related to increased age [67]. Scientific evidence also suggests that aging-related biological changes, such as degenerative changes in muscles, tendons, ligaments, and joints, may contribute to the pathogenesis of MSDs [68]. The older shopkeepers may have accumulated overtime

workload, and this cumulative effect may also explain the higher prevalence of WRMSDs among older shopkeepers than among their younger counterparts [69].

In our study analysis, there has been found a significant relationship between WRMSDs and BMI. Shopkeepers with an overweight body mass index had a 1.09 times greater prevalence of WRMSDs than those with a normal body mass index. Similar findings were found in studies conducted in India [70, 71] and Indonesia [72]. The plausible reason might be that being overweight is one of the health risk factors with the potential to aggravate the occurrence of WRMSD by increasing the physiological and mechanical load on tissues [71]. A high amount of adipose tissue around the muscles and joints due to increased BMI can also limit a person's movements, thereby stressing musculoskeletal tissues and potentially resulting in musculoskeletal pain [71, 73]. Thus, being overweight causes fatigue and postural discord, which could lead to severe MSDs [74].

Our finding indicated that the prevalence of WRMSDs was 1.32 times higher in stressed shopkeepers than in non-stressed shopkeepers. This finding is in line with the results reported in Ethiopia [34, 75], Iran [76], and India [77]. The plausible reason could be that job stress causes an increase in the release of catecholamine and cortisol hormones, which negatively impacts the structure and function of muscles, tendons, and ligaments [78]. Scientific evidence quantifies that high cortisol concentrations in the blood influence protein and carbohydrate metabolism in muscle tissue. Increased cortisol levels cause muscle weakness by increasing the release of gluconeogenesis substrates from peripheral tissues. As a result, job stress can result in increased muscle tension and blood flow restrictions, leading to WRMSDs and chronic pain [79]. In addition, individuals' behaviors might be changed under stress, resulting in a faster rate at which they complete tasks. This may lead to faster and more forceful responses during work tasks, with an increased chance of exposure to biomechanical hazards that may aggravate MSDs [80].

Prolonged static sitting is a growing occupational health risk in the workplace, and it is a risk factor for a variety of health problems [81]. Our analysis found that shopkeepers who were seated for long periods were more prone to work-related musculoskeletal complaints. Similar results were reported in other studies [22, 81]. This could be because prolonged sitting can put a strain on muscles, tendons, and ligaments, as well as raise disc pressure. This increases the risk of pain, discomfort, and injuries associated with postural stress disorders, joint compression, and soft-tissue (muscles, tendons, and ligaments) injuries [82, 83]. In addition, sitting for a long period prime to monotonously low overall energy consumption. This may lead to a situation where the body's energy demand for the back region is well below what is recommended for a healthy lifestyle. In this way, the combination of a low metabolic level and lowered blood circulation can eventually lead to muscle degeneration and osteoporosis. As well, prolonged sitting reduces the amount of oxygen that reaches muscles and organs. Therefore, extended periods of sitting have a detrimental effect on the emergence of MSDs.

## Strengths and limitations of the study

The strength of this study stems from large sample with a very good response rate (98.6%), which may allow for rigorous statistical analysis. The main limitations of this study are, firstly, that due to the cross-sectional study design, it is not possible to establish causal relationships between variables. Secondly, because of the study used a self-report assessment method (questionnaire survey), reporting bias and recall bias may have influenced the results. Finally, we didn't conduct a posture analysis to determine how much each participant had been exposed to ergonomic factors at work because of concerns about the feasibility of the large sample size used in the study; instead, we looked at potential risk factors that could lead to WRMDs.

## Conclusions

This study revealed that shopkeepers face an alarmingly high prevalence (81.1%) of WRMSDs. Being female, being older, being overweight, having job stress, and prolonged sitting in the same position were factors significantly associated with WRMSD. These findings call for the urgent development and implementation of preventive measures, including ergonomic adjustments, education and training programs, stress management techniques and the promotion of physical activity, to protect this vulnerable workforce from the debilitating effects of WRMSDs. The departments of Environmental and Occupational Health and Safety and Physiotherapy at the College of Medicine and Health Sciences, University of Gondar, Ethiopia, can address this challenge synergistically through a multi-pronged approach. By conducting ergonomic assessments, implementing awareness programs, offering targeted training programs and advocating for improved market infrastructure, these departments can empower shopkeepers to work safely and prevent musculoskeletal problems. By working with local communities, businesses and policy makers, sustainable solutions can be found to ensure the long-term health and quality of life of shopkeepers. Further research and interventions are warranted to develop comprehensive strategies and guidelines to address the high prevalence of WRMSD among shopworkers and ultimately improve their occupational health and wellbeing.

## Supporting information

**S1 File. The study questionnaire.**
(PDF)

**S1 File. Minimal anonymized data set used in analysis.**
(SAV)

## Acknowledgments

We would like to thank the University of Gondar, College of Medicine and Health Sciences and Comprehensive Specialized Hospital, and Institute of Public Health for providing ethical clearance. The authors are also very much grateful to all the data collectors, the supervisor, and the study participants.

## Author Contributions

**Conceptualization:** Amensisa Hailu Tesfaye, Gebisa Guyasa Kabito.

**Data curation:** Amensisa Hailu Tesfaye, Gebisa Guyasa Kabito, Fantu Mamo Aragaw, Tesfaye Hambisa Mekonnen.

**Formal analysis:** Amensisa Hailu Tesfaye, Gebisa Guyasa Kabito, Fantu Mamo Aragaw, Tesfaye Hambisa Mekonnen.

**Investigation:** Amensisa Hailu Tesfaye, Gebisa Guyasa Kabito, Fantu Mamo Aragaw, Tesfaye Hambisa Mekonnen.

**Methodology:** Amensisa Hailu Tesfaye, Gebisa Guyasa Kabito, Fantu Mamo Aragaw, Tesfaye Hambisa Mekonnen.

**Software:** Amensisa Hailu Tesfaye.

**Supervision:** Amensisa Hailu Tesfaye, Gebisa Guyasa Kabito, Fantu Mamo Aragaw, Tesfaye Hambisa Mekonnen.

**Validation:** Amensisa Hailu Tesfaye, Gebisa Guyasa Kabito, Fantu Mamo Aragaw, Tesfaye Hambisa Mekonnen.

**Visualization:** Amensisa Hailu Tesfaye.

**Writing – original draft:** Amensisa Hailu Tesfaye, Gebisa Guyasa Kabito, Fantu Mamo Aragaw, Tesfaye Hambisa Mekonnen.

**Writing – review & editing:** Amensisa Hailu Tesfaye, Gebisa Guyasa Kabito, Fantu Mamo Aragaw, Tesfaye Hambisa Mekonnen.

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
