## [Decision Letter · Decision Letter 0]

4 Apr 2023

PONE-D-22-30367Prevalence and risk factors of work-related musculoskeletal disorders among shopkeepers in Northwestern Ethiopia: evidence from a workplace cross-sectional studyPLOS ONE

Dear Dr. Amensisa,

Thank you for submitting your manuscript to PLOS ONE. After careful consideration, we feel that it has merit but does not fully meet PLOS ONE’s publication criteria as it currently stands. Therefore, we invite you to submit a revised version of the manuscript that addresses the points raised during the review process.

We look forward to receiving your revised manuscript.

Kind regards,

Mohammad Hayatun Nabi, MBBS, MHSM, MPH, PHD

Academic Editor

PLOS ONE

Reviewers' comments:

Reviewer's Responses to Questions

**Comments to the Author**

1. Is the manuscript technically sound, and do the data support the conclusions?

Reviewer #1: Yes

Reviewer #2: Yes

2. Has the statistical analysis been performed appropriately and rigorously? 

Reviewer #1: Yes

Reviewer #2: Yes

3. Have the authors made all data underlying the findings in their manuscript fully available?

Reviewer #1: Yes

Reviewer #2: No

4. Is the manuscript presented in an intelligible fashion and written in standard English?

Reviewer #1: No

Reviewer #2: Yes

5. Review Comments to the Author

Reviewer #1: Thank you for allowing me to review this paper. The article was nicely written and well-described by the authors. However, I have a few questions regarding the manuscript. Kindly find my comments below,

Overall:

• Grammatical mistakes evident throughout the manuscript

• Overall, the manuscript looked to be plagiarized from multiple sources. Proper references and writing patterns need to be changed. Major sources:

https://doi.org/10.1136/bmjopen-2022-069019

https://doi.org/10.1155/2021/6082506

In the abstract section,

• The descriptive portion is usually presented with numbers and percentages for the categorical variables and mean with standard deviation for the continuous variables. I suggest not using a confidence interval to describe the prevalence of a disease.

In the Background section,

• Plenty of studies have been conducted so far regarding WRMDs. Then what will be the novelty of running such a study where we have already aware of the negative impact of a sedentary lifestyle?

Methodology section

• In the calculation, authors took prevalence as 50%, where studies have been conducted in the same country on different populations or different countries on the same folks. Then why 50%?

• In total, how many questions were in the questionnaire?

• What was the average time to complete one interview?

• Were all the scales have been validated before use?

• What were the Cronbach Alpha of the individual scales?

• Were the authors included piloting data in the final analysis?

• How many data collector was involved in this research?

• The authors mentioned that participants completed the questionnaire during the face-to-face interview. But the reason for not taking informed written consent needed to be clarified. This is a serious concern. How could the authors maintain the authenticity of the data and the data collection procedures? It could be a possibility of bias.

Result section

• The authors should place age as a continuous variable under the Table 1 section with age groups.

• Numbers were mismatched in the "Education Level" variable with the explanation.

• "Three-fourths of the shopkeepers (74.4%) had a low perceived disability of musculoskeletal pain"- should be changed to "Around three-fourths of the shopkeepers (74.4%) had a low perceived disability of musculoskeletal pain".

• What is the justification for using table 4? Why AOR? What were the other variables adjusted here? The authors should exclude that, as it has already been presented in table 6.

Discussion section:

• What is the logic behind evaluating the WRMSDs in the past 12 months? There could be a high possibility of recall biases here.

• In the discussion part, the authors stated, "Besides, measures to prevent injuries and safeguard the health and safety of workers are often lean in many developing nations". Kindly provide some solid references for this statement.

Suggestion: The study questionnaire could be attached to the manuscript as a supplementary.

Reviewer #2: I have read the "Prevalence and risk factors of work-related musculoskeletal disorders among shopkeepers in Northwestern Ethiopia: evidence from a workplace cross-sectional study" manuscript by Amensisa eta al. The article is well written.

Minor comments

line 10: corresponding author/ correspondence not "correspondence author"

Abstract

95% CI of the prevalence does not give sense in the abstract section. Authors may mention this either in the result and preferably in the discussion section as it helps to compare with existing literature. In the abstract, the point prevalence is sufficient.

Background is well written.

methods

under study design and period no need to rewrite the objective as it does not add any value.

in the study setting and area unnecessary details are presented and important information about the total number of shoekeepers is missed.

Line 261 is similar to lines 262 to 263

line 295: What is valid questionnaire?

Major issues

Lines 138 to 140:

Confounding can be addressed by restriction, matching and multivariable analysis. Why do authors prefer restriction as a method of choice and what impacts does this method have on the study's external validity?

Authors claimed that they excluded pregnant women. How did they check pregnancy?

Measurement of variables

1. what was the pre-test result?

2. was NMQ validated in Ethiopia?

3. Nothing was said about whether the tools used were validated in Ethiopia.

4. line 249 "reduced number of questions"? What were the items deleted?

What was the the α coefficient of 0.76 for? Authors have used a number of tools and it seems trivial to report a single α coefficient for divergent constructs which measure quite different concepts.

Line 268, why logistic regression as a method and odds ratio as effect size? As the authors have reported 81.1% prevalence, it would have been good if they use log binomial and report prevalence ratio instead of odds ratio. As it stands now the OR over estimates the PR and the way authors interpreted the odds is not correct. Authors may wish to perform re analysis using log binomial and report prevalence ratio as effect size measure or atleast correct the way they interpreted the odds and mention the limitation of using OR over PR for high prevalence report in the limitation section if they wish to stick on logistic regression coming up with convincing reason.

The authors may wish to annex the ethical clearance paper.

The justification for skipping written consent and going to verbal consent is not convincing

Physical activity was mentioned as yes/no in the operational definition and reported with frequencies in the result part. how?

The interpretation of odds ratio as relative risk in cross sectional study is trivial.

Authors mentioned that females may have additional strain at home. How did the authors single out whether the MSDS was from exposure to their work or any other source?

Authors used chronic disease as yes/no? how was it ascertained?

line 407 "Accordingly, the likelihood of developing WRMSDs increases with age" To say so, authors were able to use the age score as continuous variable. What was the base of classification for age, work experience and family size?

Good luck!

6. PLOS authors have the option to publish the peer review history of their article (what does this mean?). If published, this will include your full peer review and any attached files.

Reviewer #1: No

Reviewer #2: No

---

## [Author Response · Author response to Decision Letter 0]

19 Apr 2023

We are very grateful for receiving the editor’s and reviewers’ comments on our manuscript, which we believe will improve our manuscript. Following the reviewers' and editors' valuable comments and recommendations, we have revised the work, and we hereby submit the revised work for your consideration along with a point-by-point response to the editor and reviewers.

Kindest regards,

Authors

---

## [Decision Letter · Decision Letter 1]

26 Oct 2023

PONE-D-22-30367R1Prevalence and risk factors of work-related musculoskeletal disorders among shopkeepers in Northwestern Ethiopia: evidence from a workplace cross-sectional studyPLOS ONE

Dear Dr. Tesfaye,

Thank you for submitting your manuscript to PLOS ONE. After careful consideration, we feel that it has merit but does not fully meet PLOS ONE’s publication criteria as it currently stands. Therefore, we invite you to submit a revised version of the manuscript that addresses the points raised during the review process.

Please see the comments from two reviewers below and in the attachment. Please note that the original Editor and reviewers became unavailable, and thus, two new reviewers were sought. Please disregard reviewer 2's suggestion to replace "sex" with "gender" if the biological meaning is intended.

We look forward to receiving your revised manuscript.

Kind regards,

Hanna Landenmark

Staff Editor

PLOS ONE

Journal Requirements:

Reviewers' comments:

Reviewer's Responses to Questions

**Comments to the Author**

1. If the authors have adequately addressed your comments raised in a previous round of review and you feel that this manuscript is now acceptable for publication, you may indicate that here to bypass the “Comments to the Author” section, enter your conflict of interest statement in the “Confidential to Editor” section, and submit your "Accept" recommendation.

Reviewer #3: (No Response)

Reviewer #4: All comments have been addressed

2. Is the manuscript technically sound, and do the data support the conclusions?

Reviewer #3: Yes

Reviewer #4: Yes

3. Has the statistical analysis been performed appropriately and rigorously? 

Reviewer #3: Yes

Reviewer #4: Yes

4. Have the authors made all data underlying the findings in their manuscript fully available?

Reviewer #3: Yes

Reviewer #4: Yes

5. Is the manuscript presented in an intelligible fashion and written in standard English?

Reviewer #3: No

Reviewer #4: Yes

6. Review Comments to the Author

Reviewer #3: Dear Author,

I am pleased to evaluate this manuscript entitled “Prevalence and risk factors of work-related musculoskeletal disorders among shopkeepers in Northwestern Ethiopia: evidence from a workplace cross-sectional study”.

The manuscript raises an important area of concern. The manuscript is good in terms of methodology. However, it should be presented in a better way using concise and standard language. Please look into the PLOS ONE manuscript preparation guide and similar articles published in the journal.

The comments and questions are forwarded in the comment boxes.

Good luck!

Reviewer #4: The revised version of research paper entitled “Prevalence and risk factors of work-related musculoskeletal disorders among shopkeepers in Northwestern Ethiopia: evidence from a workplace cross-sectional study” is a good work conducted by the authors. The manuscript needs to incorporate the following comments/suggestions as:

1. In the text author must quote reference carefully, e.g. in lines 102-103, after reference number 26, the next reference number will be 27 not 31? Authors are suggested to re-numbered the references accordingly both in main text and in reference section.

2. In line 146, write as “Maraki and Arada”.

3. In line 172, write as “Alcohol intake”.

4. Rewrite the lines 305-306, are authors explaining their results? If yes the how their results are in accordance with WHO classification? Also add reference support of WHO classification.

5. In lines 352 and 353 in kg/m2, “2” will be in superscript font.

6. In reference section, line 518, 2022 appears twice.

7. In lines 531 and 532 w, h and o in world health organization will be in capital font.

8. Authors must write the references as per “PLOS one” style.

9. Reference no 25 is not complete.

10. Add page no in reference no 33.

11. In line 605, year 2020 appears twice.

12. Reference no 83 is not complete.

13. In Table 1 instead of using word “Sex” better to use “Gender”.

14. In Table 2 check the total of percentage of BMI and Chronic illness reported, it’s not 100 %?

7. PLOS authors have the option to publish the peer review history of their article (what does this mean?). If published, this will include your full peer review and any attached files.

Reviewer #3: **Yes: **Abebaw Jember Ferede

Reviewer #4: **Yes: **Dr. Hardeep Rai Sharma, Institute of Environmental Studies, Kurukshetra University, Kurukshetra, Haryana, India

---

## [Author Response · Author response to Decision Letter 1]

20 Nov 2023

We are very grateful for receiving the reviewers’ comments on our manuscript, which we believe will improve our manuscript. Following the reviewers' valuable comments and recommendations, we have revised the work, and we hereby submit the revised work for your consideration along with a point-by-point response to the reviewers.

Thank you all for your constructive and valuable comments and concerns.

Note that all changes to the manuscript are included in the track change too.

Kindest regards,

Authors

---

## [Decision Letter · Decision Letter 2]

11 Dec 2023

PONE-D-22-30367R2Prevalence and risk factors of work-related musculoskeletal disorders among shopkeepers in Northwestern Ethiopia: evidence from a workplace cross-sectional studyPLOS ONE

Dear Dr. Tesfaye,

Thank you for submitting your manuscript to PLOS ONE. After careful consideration, we feel that it has merit but does not fully meet PLOS ONE’s publication criteria as it currently stands. Therefore, we invite you to submit a revised version of the manuscript that addresses the points raised during the review process.

**ACADEMIC EDITOR: Please insert comments here and delete this placeholder text when finished.** Be sure to: Refer to the reviewers comments below and ensure you act on each comments prior to submitting the updated version. 

We look forward to receiving your revised manuscript.

Kind regards,

Haruna Musa Moda

Academic Editor

PLOS ONE

Journal Requirements:

Reviewers' comments:

Reviewer's Responses to Questions

**Comments to the Author**

1. If the authors have adequately addressed your comments raised in a previous round of review and you feel that this manuscript is now acceptable for publication, you may indicate that here to bypass the “Comments to the Author” section, enter your conflict of interest statement in the “Confidential to Editor” section, and submit your "Accept" recommendation.

Reviewer #3: All comments have been addressed

Reviewer #4: (No Response)

2. Is the manuscript technically sound, and do the data support the conclusions?

Reviewer #3: Partly

Reviewer #4: Yes

3. Has the statistical analysis been performed appropriately and rigorously? 

Reviewer #3: No

Reviewer #4: Yes

4. Have the authors made all data underlying the findings in their manuscript fully available?

Reviewer #3: Yes

Reviewer #4: Yes

5. Is the manuscript presented in an intelligible fashion and written in standard English?

Reviewer #3: No

Reviewer #4: Yes

6. Review Comments to the Author

Reviewer #3: It is recommended that manuscripts are presented written in standardized English. It is also important to follow the journal's author guide to improve the quality of the manuscript. In your rebuttal concerning the write up, it is important to reduce repetitiveness of texts in the manuscript. For instance, presenting the findings in text and in table, presenting similar ideas in a paragraph also should be avoided.

Reviewer #4: The re-revised version of the research paper entitled “Prevalence and risk factors of work-related musculoskeletal disorders among shopkeepers in Northwestern Ethiopia: evidence from a workplace cross-sectional study” is an appreciable efforts of the authors. Still, the manuscript needs to incorporate the following comments/suggestions as:

1. As suggested earlier, authors must write the reference carefully and strictly as per “PLOS ONE” style, still many references in the reference section needs careful editing e.g. reference number 8 (correct world health organization); no. 9-13, 16, 18, 24, 36-38, 43, 44, 47-49, 52, 53, 55, 59, 60, 63-65, 67, 68, 73, 74, 79 and 82 (write journal names properly); no. 15 (correct Burden of Disease); no. 80 (needs proper quoting) etc. Further many journal names are written full while other in abbreviations? Ref. no 25, 32, 80 and 83 are not complete add either their websites or other details. In ref no. 51 add country name after Sebelas Maret University. Authors are suggested to do changes in different font color in the revised R3 manuscript.

2. As suggested earlier, In Table 1 instead of using word “Sex” better to use “Gender”, what is authors opinion about the change?

3. The overall sample size for this study was 634 (lines 144 and Table 1) but authors analyzed 625 questionnaires (lines 288) what was the reason for 98.6% response rate can be briefly added in Socio-demographic characteristics of study participants section.

4. Author’s recommendations in lines 446-450 are OK but some sort of intervention or solution can be expected from Environmental and Occupational and Physiotherapy departments in the College of Medicine and Health Science of the University of Gondar, Ethiopia! Authors can suggests some basic exercises and correct sitting postures to the studied shopkeepers for a time period so that the purpose of “Community Service” for health professionals can be achieved.

7. PLOS authors have the option to publish the peer review history of their article (what does this mean?). If published, this will include your full peer review and any attached files.

Reviewer #3: No

Reviewer #4: **Yes: **Dr. HARDEEP RAI SHARMA

---

## [Author Response · Author response to Decision Letter 2]

14 Jan 2024

We are very grateful for receiving the editor’s and reviewers’ comments on our manuscript, which we believe will improve our manuscript. Following the reviewers' and editors' valuable comments and recommendations, we have revised the work, and we hereby submit the revised work for your consideration along with a point-by-point response to the editor and reviewers.

Kindest regards,

Authors

---

## [Decision Letter · Decision Letter 3]

31 Jan 2024

PONE-D-22-30367R3Prevalence and risk factors of work-related musculoskeletal disorders among shopkeepers in Ethiopia: Evidence from a workplace cross-sectional studyPLOS ONE

Dear Dr. Tesfaye,

Thank you for submitting your manuscript to PLOS ONE. After careful consideration, we feel that it has merit but does not fully meet PLOS ONE’s publication criteria as it currently stands. Therefore, we invite you to submit a revised version of the manuscript that addresses the points raised during the review process.**May I once more request you have a closer look at the 3rd reviewers comments based on the revised work you have sent and respond to this accordingly. **Please submit your revised manuscript by Mar 16 2024 11:59PM. If you will need more time than this to complete your revisions, please reply to this message or contact the journal office at plosone@plos.org. Please include the following items when submitting your revised manuscript:A rebuttal letter that responds to each point raised by the academic editor and reviewer(s). You should upload this letter as a separate file labeled 'Response to Reviewers'.A marked-up copy of your manuscript that highlights changes made to the original version. You should upload this as a separate file labeled 'Revised Manuscript with Track Changes'.An unmarked version of your revised paper without tracked changes. You should upload this as a separate file labeled 'Manuscript'.If applicable, we recommend that you deposit your laboratory protocols in protocols.io to enhance the reproducibility of your results. Protocols.io assigns your protocol its own identifier (DOI) so that it can be cited independently in the future. For instructions see: https://journals.plos.org/plosone/s/submission-guidelines#loc-laboratory-protocols. Additionally, PLOS ONE offers an option for publishing peer-reviewed Lab Protocol articles, which describe protocols hosted on protocols.io. Read more information on sharing protocols at https://plos.org/protocols?utm_medium=editorial-email&utm_source=authorletters&utm_campaign=protocols.

We look forward to receiving your revised manuscript.

Kind regards,

Haruna Musa Moda

Academic Editor

PLOS ONE

Journal Requirements:

Reviewers' comments:

Reviewer's Responses to Questions

**Comments to the Author**

1. If the authors have adequately addressed your comments raised in a previous round of review and you feel that this manuscript is now acceptable for publication, you may indicate that here to bypass the “Comments to the Author” section, enter your conflict of interest statement in the “Confidential to Editor” section, and submit your "Accept" recommendation.

Reviewer #3: (No Response)

Reviewer #4: All comments have been addressed

2. Is the manuscript technically sound, and do the data support the conclusions?

Reviewer #3: Partly

Reviewer #4: Yes

3. Has the statistical analysis been performed appropriately and rigorously? 

Reviewer #3: No

Reviewer #4: Yes

4. Have the authors made all data underlying the findings in their manuscript fully available?

Reviewer #3: Yes

Reviewer #4: Yes

5. Is the manuscript presented in an intelligible fashion and written in standard English?

Reviewer #3: No

Reviewer #4: Yes

6. Review Comments to the Author

Reviewer #3: Dear Author,

I am pleased to evaluate this manuscript entitled “Prevalence and risk factors of work-related musculoskeletal disorders among shopkeepers in Northwestern Ethiopia: evidence from a workplace cross-sectional study”.

The manuscript highlights an important area of concern and has a good methodology. However, it would be beneficial to present the information in a more concise and standard language. I recommend consulting the PLOS ONE manuscript preparation guide and other articles published in the journal for guidance on how to improve the presentation of your work.

Based on the nature of the outcome variable, I recommend reanalyzing the data using the appropriate statistical analysis method.

The comments and questions are forwarded in the comment boxes.

Good luck!

Reviewer #4: A good research work which can be repeated after implementing recommendations. Intervention in always an important part of research. All the best

7. PLOS authors have the option to publish the peer review history of their article (what does this mean?). If published, this will include your full peer review and any attached files.

Reviewer #3: No

Reviewer #4: **Yes: **Dr HARDEEP RAI SHARMA

---

## [Author Response · Author response to Decision Letter 3]

7 Feb 2024

We gratefully acknowledge every one of your insightful and incredibly intriguing remarks.

We look forward to your positive response as soon as possible.

---

## [Decision Letter · Decision Letter 4]

7 Mar 2024

Prevalence and risk factors of work-related musculoskeletal disorders among shopkeepers in Ethiopia: Evidence from a workplace cross-sectional study

PONE-D-22-30367R4

Dear Dr. Tesfaye

We’re pleased to inform you that your manuscript has been judged scientifically suitable for publication and will be formally accepted for publication once it meets all outstanding technical requirements.

Kind regards,

Haruna Musa Moda

Academic Editor

PLOS ONE

Additional Editor Comments (optional):

Reviewers' comments:

Reviewer's Responses to Questions

**Comments to the Author**

1. If the authors have adequately addressed your comments raised in a previous round of review and you feel that this manuscript is now acceptable for publication, you may indicate that here to bypass the “Comments to the Author” section, enter your conflict of interest statement in the “Confidential to Editor” section, and submit your "Accept" recommendation.

Reviewer #4: All comments have been addressed

Reviewer #5: All comments have been addressed

2. Is the manuscript technically sound, and do the data support the conclusions?

Reviewer #4: Yes

Reviewer #5: Yes

3. Has the statistical analysis been performed appropriately and rigorously? 

Reviewer #4: Yes

Reviewer #5: Yes

4. Have the authors made all data underlying the findings in their manuscript fully available?

Reviewer #4: Yes

Reviewer #5: Yes

5. Is the manuscript presented in an intelligible fashion and written in standard English?

Reviewer #4: Yes

Reviewer #5: Yes

6. Review Comments to the Author

Reviewer #4: The revised version of the manuscript in response to the reviewers comments improves the quality of the work. In my opinion as authors incorporates the reviewers comments and suggestions is appreciable.

Reviewer #5: The manuscript is well written and is in a state that could be accepted for publication.

Comments to be addressed by the authors during proof reading

Line 758 - Table 1 – Stick to one decimal or two decimal place to maintain uniformity. The below errors could be due to the decimal confusion.

Line 761 – Table 2 – Under physical exercise, the percentage does not add up to 100

Line 761 – Table 2 – Chronic illness reported, the percentage does not add up to 12.8

Line 763 – Table 3 – Type of Shop, the percentage is 100.1.

Please check all the tables for this small glitch

7. PLOS authors have the option to publish the peer review history of their article (what does this mean?). If published, this will include your full peer review and any attached files.

Reviewer #4: **Yes: **HARDEEP RAI SHARMA

Reviewer #5: **Yes: **Ravi Rangarajan

---

## [Editor Report · Acceptance letter]

11 Mar 2024

PONE-D-22-30367R4 

PLOS ONE

Dear Dr. Tesfaye, 

I'm pleased to inform you that your manuscript has been deemed suitable for publication in PLOS ONE. Congratulations! Your manuscript is now being handed over to our production team.

Kind regards, 

on behalf of

Dr. Haruna Musa Moda 

Academic Editor

PLOS ONE